# Yeast Telomerase RNA Flexibly Scaffolds Protein Subunits: Results and Repercussions

**DOI:** 10.3390/molecules25122750

**Published:** 2020-06-14

**Authors:** David C. Zappulla

**Affiliations:** Department of Biological Sciences, Lehigh University, Bethlehem, PA 18015, USA; david.zappulla@lehigh.edu; Tel.: +1-610-758-5088

**Keywords:** telomerase, RNA, lncRNA, RNP, flexible scaffold, scaffold, noncoding RNA, ncRNA, TLC1, telomere, senescence

## Abstract

It is said that “hindsight is 20-20,” so, given the current year, it is an opportune time to review and learn from experiences studying long noncoding RNAs. Investigation of the *Saccharomyces cerevisiae* telomerase RNA, TLC1, has unveiled striking flexibility in terms of both structural and functional features. Results support the “flexible scaffold” hypothesis for this 1157-nt telomerase RNA. This model describes TLC1 acting as a tether for holoenzyme protein subunits, and it also may apply to a plethora of RNAs beyond telomerase, such as types of lncRNAs. In this short perspective review, I summarize findings from studying the large yeast telomerase ribonucleoprotein (RNP) complex in the hope that this hindsight will sharpen foresight as so many of us seek to mechanistically understand noncoding RNA molecules from vast transcriptomes.

## 1. Introduction

Scientists, like sailors, know that only a fraction of what actually exists in this world is currently within one’s “arc of visibility” (i.e., what is visible to the horizon at a given time). In terms of the physical organization of biological macromolecules and how they behave in space and time, what we can see and understand at this point is based largely on highly ordered matter. This is a consequence of methods such as X-ray crystallography and cryo-EM being our main visualization tools. These approaches are magnificently powerful, but they also only produce intelligible pictures if the purified target is structured (or can be made to be). Nevertheless, biologists ultimately aim to perceive what is beyond the proverbial horizon in terms of the molecules of life, and this requires conceiving of an expanse of physically disordered macromolecular complexes that are currently beyond our vantage. In this perspective, I briefly review results that have demonstrated pronounced organizational flexibility of the large telomerase RNA-protein complex, and then extrapolate what these findings may mean about the nature and extent of flexible RNPs possibly residing outside our current “arc.”.

Life evolves based on DNA mutations that lead to fitness improvements. Genes are expressed as RNA and, in turn, frequently also as protein. If an evolutionary advantage for an organism would be obtained by building a highly flexible molecular complex based on one of these two gene-encoded polymers, one would be apt to argue that RNA provides more potential. This is because of (1) RNA’s larger size (per residue, and in terms of secondary structure), and (2) RNA’s chemical composition, with its higher solubility in aqueous cellular environments than proteins, as evidenced by what happens in a phenol extraction of a cell lysate. Life on earth began with RNA [1], and the fact that RNA evolved as the initial biochemical building block in itself should also tell us something about the scope of its capabilities. Furthermore, the wide variety of transcribed long noncoding RNAs (lncRNAs) that comprise the “dark matter” of eukaryotic transcriptomes could provide a diverse and useful array of flexible scaffolds on which proteins and other highly structured molecules may swing and slide to execute fundamental molecular biological processes.

The nature and extent of flexibility of RNA in RNP complexes has been highlighted in the last two decades by studies of the telomerase RNP enzyme in the yeast *S. cerevisiae*. The telomerase RNP is required by most eukaryotes to address the inadequacy of DNA polymerases to entirely finish copying chromosome ends [2]. Structure–function studies of yeast telomerase reverse-transcriptase RNP commenced around the year 2000, once both the telomerase RNA (TLC1) and the reverse transcriptase protein subunit (TERT; Est2 in yeast) had been identified [3,4]. Other subunits of yeast telomerase beyond the TLC1•TERT core enzyme had been found previously, and some of these are also essential for telomerase in vivo to avoid cellular senescence. Among these other subunits are the critical Est1 protein (the first telomerase-subunit gene identified) [5,6], the Ku70/80 heterodimer [7,8] (also known for its central role in DNA end-joining), and the Sm_7_ heteroheptamer [9] (which also binds snRNAs of the spliceosome).

In the following section of this perspective, I first synopsize the results that we have obtained from investigating *S. cerevisiae* telomerase RNP structure and organization that led us to the model that the large TLC1 RNA flexibly scaffolds the holoenzyme’s protein subunits. In Section 3, I then turn to discussing what the flexible scaffold model may imply for other large RNPs.

## 2. Results from Investigating Yeast Telomerase RNA

### 2.1. Rapid Evolution

As the genomes of budding yeast species were sequenced in the years following the completion of *Saccharomyces cerevisiae* in 1997, related telomerase RNA genes were identified and aligned. Telomerase RNA gene identification is very difficult, since these sequences evolve so fast that even distantly related fungi have no discernible sequence identity. The identification of multiple *Saccharomyces* and *Kluyveromyces* species’ telomerase RNAs led to a flurry of initial insights about conserved local secondary structure elements and their protein-binding partners. Articles reporting these discoveries showed that these other budding yeast telomerase RNAs are also >1 kb in length, and are divergent in primary structure (i.e., sequence) [6,10,11,12]. Even within the genus *Saccharomyces*, overall nucleotide sequence identity is, strikingly, less than 50% [11,13,14]. The rate of evolution of telomerase RNA sequences is far faster than other essential noncoding RNAs, such as the ribosome, RNase P, and snRNAs, at 80–99% identity [13]. Nevertheless, there are multiple patches of well-conserved sequences scattered across the length of the large telomerase RNAs [11].

### 2.2. Identifying S. cerevisiae Telomerase RNA Secondary Structure

The identification of budding yeast telomerase RNA sequences provided the starting point for next determining secondary structure. But the RNAs were very large, and so rapidly evolving in sequence and length that it made alignment in some sections rather difficult. The large RNAs and rapid evolution challenged computational modeling and identification of covarying nucleotides that could help solve the secondary structure. In 2002, the Cech lab reported that large truncations could be made in the *S. cerevisiae TLC1* gene without causing senescence, whereas central conserved segments were essential, and tended to overlap with regions required for binding to the Est2/TERT and Est1 essential subunits [15]. At this time, a question was emerging: why was so much of yeast telomerase RNA evolving as if it were a form of “junk” RNA?

In 2004, secondary structure models for TLC1 were reported (Figure 1). The two models were nearly identical and were independently derived from (1) bioinformatic modeling of lowest-free-energy RNA conformations and (2) alignments of the TLC1 sequences from several *Saccharomyces* species to identify covarying nucleotides [13,14]. It was rather heretical to consider that an *Mfold* software prediction of a large RNA could be accurate (although the software does tend to correctly predict an average of ~70% base pairings [16]). Nevertheless, I generated *Mfold* secondary structure predictions for the range of truncated TLC1 sequences published by Livengood and Cech (2002) [13] while I was postdoc in the Cech lab, and noticed an essentially perfect correlation between the major differences between these TLC1 alleles’ *Mfold* predictions compared to wild type and the severity of the observed phenotypes. However, the *Mfold* lowest-free-energy modeling required some important constraints to accommodate covarying nucleotides; these supported the existence of some long-range base pairings and helices not present in the lowest-free-energy structure computed by *Mfold*. These covarying base pairs I identified with help from *RNAalifold* [17]. Primarily, these constraints were necessary for correctly modeling the core-enclosing helix and the overall structure of the highly branched multi-way junction at the central core’s hub (Figure 1). In other regions of TLC1, the covarying nucleotides were independently predicted to be paired in the *Mfold* energetics-driven modeling, adding further confidence and encouragement that the data were leading towards a compelling consensus. The model ultimately published by us [13] and another by the Wellinger group [14] were essentially identical, adding yet further confirmation.

Overall, the TLC1 secondary structure has three major, mostly helical RNA arms that radiate away from a central core, containing the template for reverse transcription, and the segments of the RNA required to bind to TERT [13,14] (Figure 1). Near the tip of each arm is a binding site for a holoenzyme subunit: either Est1, Ku, or Sm_7_. (Since then, Pop1/6/7 have also been shown to bind the RNA near Est1 [27,28,31,32].) We found that this organization of the large, rapidly evolving, truncation-tolerant yeast telomerase RNA supported our hypothesis that it functions as a flexible scaffold for these arm-bound subunits. As unusual as the flexible scaffold hypothesis was at the time, it did fit with the modeling and experimental results.

### 2.3. Functional Repositioning of Holoenzyme Subunits and the Ends of TLC1

Encouraged by the accumulating evidence supporting the hypothesis that yeast telomerase RNA is essentially a tether for the holoenzyme protein subunits, we set out to test this in vivo. If telomerase RNP had a physical organization that did not require precise positioning of RNA-bound protein subunits in 3D space, then if we relocated, for example, the essential Est1 protein’s binding site to other arms, hundreds of bases away from its natural location, it should theoretically still perform its function in recruiting telomerase to chromosome ends and also activating it [33,34]. Indeed, we found that this was the case: in all three new locations tested (positions 220, 450 and 1033), the essential Est1 protein provided its function in telomerase and allowed cells to grow well and avoid senescence [13] (Figure 1, orange arrows).

#### 2.3.1. Est1

The fact that an essential protein subunit of telomerase could be moved to diverse locations on the RNA secondary structure, as mentioned above, is particularly significant when one considers how different this result would be in other essential enzymatic RNPs. If one relocated a binding site in an rRNA for a protein subunit to a location distant from its native position, that subunit (and potentially the ribosome) would almost certainly not function. Thus, it was probable that telomerase has a different physical organization than other well-studied RNP complexes.

#### 2.3.2. Ku

The Ku70/80 heterodimer is well known for its key role in DNA repair, but it also binds to telomerase RNA in yeast and also in humans [7,35]. In yeast, it is clear that Ku recruits telomerase to chromosome ends by binding to the transcriptional silencing factor Sir4 [36,37]. The evidence mounted further for the TLC1 flexible scaffold hypothesis when we tested whether the Ku heterodimer’s binding position in TLC1 could also be moved with retention of function. Indeed, it did; moving the Ku site supported an increase in telomere length compared to a Ku-binding-defective allele, and even more clearly with Ku-binding-defective sites at the same positions [23].

#### 2.3.3. Sm_7_ and the 5′ and 3′ ends

The Sm_7_ complex, which binds to a consensus sequence near the 3′ end of TLC1 [9], can also function at alternate locations in Mini-T and full-length TLC1 [24,29]. One additional facet of these experiments was that the 5′ and 3′ ends of TLC1 needed to be co-relocated (i.e., the RNA was circularly permuted) along with the Sm_7_ subunit in these alleles, since Sm_7_ binding location participates in the proper processing of the 3′ terminus. Thus, since these circularly permuted Sm_7_-repositioned TLC1 constructs were functional, TLC1 tolerates relocation of its termini—a feature of the RNA that begets a different co-transcriptional folding pathway, due to the altered order in which the RNA polymer is synthesized, in addition to causing local structural changes at the new end sites.

In summary, with the above relocations of holoenzyme-specific subunits, including the essential Est1, and important Ku and Sm_7_ complex components, we showed that these subunit-binding sites are functional modules, and likely all represent structured domains, in the context of the RNP as a whole. In fact, the crystal structure of yeast Ku bound to the TLC1 hairpin was recently solved, showing details of this particularly discrete module [38]. TLC1 RNA provides a flexible scaffold for these subunit modules, presumably since they need to function in cis on the RNP, but clearly with multiple locations at which each can work.

### 2.4. Deleting or Stiffening the Presumably Pliable TLC1 RNA Arms

Given that subunits’ binding sites can be repositioned on the yeast telomerase RNA with retention of their function, demonstrating that the holoenzyme subunits are modular in nature, to what extent does this show that they are orchestrated by TLC1 in a physically flexible, or pliable, manner in wild-type telomerase? In terms of what we know of TLC1 secondary structure, since the overall long-armed, Y-shaped molecule (Figure 1) is littered with bulges and loops [13], it certainly looks as though it would be a floppy tether. A recent atomic force microscopy study in physiological conditions showed that junctions such as this in nucleic acids can allow 100° freedom, for example [39]. We performed two different experiments that directly address the question of physical pliability:We deleted the bulk of the three rapidly evolving RNA arms to test whether they were important [25]. These 384–500-nt “Mini-T” RNAs still contained all of the conserved patches of sequences that bind to proteins; they only have the rapidly evolving medial segments of the RNA arms excised (Figure 1).We stiffened the arms by converting them all (and each single- and double-arm combination) to perfect double-stranded RNA (dsRNA) rod-like struts by removing all of the bulges and loops that connect all the dsRNA segments [26] (Figure 1). The triple-stiff-arm RNA was called TSA-T. As with Mini-T, TSA-T did not have altered structure in the conserved patches that we knew bound to protein subunits. (N.B., long, perfect stretches of dsRNA do not feed into a siRNA pathway in *S. cerevisiae* cells, since the organism lacks miRNA-processing components.)

Mini-T and TSA-T RNAs were functional, supporting sufficient telomerase activity in cells to maintain stable telomeres and prevent senescence. However, the fitness of a truncated-arm construct, Mini-T(500), was tested quantitatively and shown to be reduced, and all of the Mini-T constructs supported very short telomeres [25].

Perhaps not surprising for Mini-T, due to having two-thirds of the natural telomerase RNA nucleotides removed, and therefore experiencing a simplified folding landscape (and avoiding misfolding [40]), these constructs performed better than TLC1 in vitro [25]. Wild-type TLC1 is nearly completely inactive when reconstituted with TERT in vitro, yet Mini-T RNAs provided robust activity for yeast telomerase [25]. This then opened the door for biochemistry experiments to determine the basic function of proteins such as the telomeric DNA end-binding protein, Cdc13, which we showed inhibits telomerase, akin to the human telomere end-binding protein, Pot1 [41,42].

Some things that we learned collectively from Mini-T and TSA-T about the flexible arms of the wild-type TLC1 RNA in yeast include (1) the wild-type arms are required for full fitness of the cells, (2) they are not essential for basic telomerase maintenance of shorter but stable telomeres, and (3) even in the case where the arms can no longer physically bend due to excision of all of the hinges (loops and bulges), telomerase RNA still functions, even in vivo. This last result supports the mounting evidence that Est1, Ku, and Sm_7_ function as modules that do not need to bind to each other or to the catalytic core for their function even in cis. Furthermore, these essential and important modules flexibly scaffolded by TLC1 are also unlikely to coordinate with each other *in trans*, since *S. cerevisiae* telomerase has been shown to function as a monomer [43].

### 2.5. TLC1 RNA Has Some Important Secondary and Tertiary Structure within Subunit-Binding Modules

Approximately 40% of TLC1 RNA comprises well-conserved patches of sequence, based on alignments with closely related budding yeast species. Within these patches (see non-bracketed RNA sections in Figure 1), there are some covarying nucleotides that do appear to reproducibly form base pairs in vivo and in vitro [6,13,14,38,44,45,46], showing that these regions’ secondary structures are also conserved, at least amongst closely related yeasts. This is not surprising, given that these regions make specific contacts with protein subunits. Furthermore, in the catalytic core, where TLC1 interacts with TERT [15,20,30], there are specific secondary structures that are nearly universally conserved among telomerase RNAs, including ciliates [47], from which telomerase was first isolated [2], and mammals [48], where telomerase plays a critical role in aging and cancer [49,50,51].

Examples of conserved regions in TLC1 are in the Est1 arm, where there is a Second Essential Est1-arm Domain (SEED) [45] or CS2a region [11,31] shown to bind to Pop1/6/7 proteins [28], and in the core, where there is an Area of Required Connectivity (ARC) [29] and core-enclosing helix (CEH) [30] (Figure 1). The conserved core region of TLC1 is about the size of the ciliate telomerase RNAs (~150 nts), and this is consistent with these streamlined telomerase RNAs/RNPs (needed in vast quantities in macronuclei) being amenable to structural biology studies using cryo-EM [52]. Interestingly, Mini-T RNAs are about the size of human telomerase RNA, hTR (451 nts), and hTR has “hypervariable regions” identified from the very beginning of vertebrate telomerase RNA secondary structure reports [53].

Thus, the fraction of 1157-nt yeast telomerase RNA that is conserved among budding yeasts essentially represents the portion of TLC1 that is shared with the Mini-T-sized 451-nt human telomerase RNA, and this is greater than the “Micro-T”-size 150-nt ciliate (catalytic core only) telomerase RNAs [19,25]. Our SHAPE chemical probing in vitro of the conserved core and Est1-arm region of TLC1 provides further evidence that these regions are indeed largely structured domains [18,45]. Therefore, TLC1 RNA provides more than a template for reverse transcription and a flexible scaffold: it also has some more “conventional,” and certainly fascinating, structured elements that provide beneficial and sometimes also essential functions. In the case of the holoenzyme-specific protein subunits (e.g., Est1, Ku, Sm_7_) of yeast telomerase RNPs, these modules have been shown to be flexibly scaffolded as reviewed in sections above, whereas, in the case of the catalytic core, there is some very interesting and impressive flexibility [29,30], yet it also has very clear limitations [29,30], consistent with the enzymatic coordination with the TERT protein and the DNA substrate.

In theory, the structured modules that comprise the minority of the overall TLC1 RNA molecule could come together in 3D space on a large flexible-scaffolding RNA to ultimately form a holoenzyme RNP with overall precisely coordinated positioning. This is important to consider and test. The data acquired thus far do not support this possibility, however. First, as mentioned above, the triple-stiff-arm allele, TSA-T is functional in vivo despite the holoenzyme-binding modules being held apart by rigidified rod-like arms that would prevent the Est1, Ku, and Sm_7_-binding regions from coming together in cis. Second, Mini-T is functional despite its presumably pliable long arms being excised, and these truncations thus “reel in” the holoenzyme subunits in towards, and nearly abutting, the catalytic core. This would confine the amount of physical sampling of these subunits in space relative to the RNP as a whole. Thus, the most compelling and parsimonious view is that holoenzyme subunits function quite independently, with the TLC1 simply flexibly tethering these modules to the RNP complex.

## 3. Repercussions

What lessons can be learned from what we have discovered so far about the large yeast telomerase RNA’s flexible scaffolding of holoenzyme subunit proteins in this RNP? The repercussions of the studies summarized above should be considered in the context of the broad range of conditions where RNA may serve to scaffold a higher-order complex. One context is simply with respect to other telomerase RNPs. A second is extrapolating to other RNPs in cells, while considering that RNA can serve as a flexible scaffold for proteins in an RNP, even one that is enzymatic. We can use the case of the TLC1 flexible-scaffold RNA paradigm when considering other RNPs. How flexible are the best-characterized RNP complexes, including those with enzymatic functions? Furthermore, with respect to what is “beyond our visible horizon” in terms of flexible RNPs, is yeast telomerase a harbinger of many more that are to be found next, such as from the class of RNAs currently referred to as long noncoding RNAs?

### 3.1. A Spectrum of RNP Flexibility

It is useful to conceive of the spectrum of physical and functional flexibility as it pertains to RNA-protein complexes. A prior review by Tom Cech and me [54] covered many well-known and more newly identified RNPs and whether they are likely to be flexibly scaffolded by RNA. Here, I focused largely on the 13 years of progress on studying yeast telomerase RNP since our prior review [54], and propose that RNA biologists endeavor to consider the extent of flexibility of known and novel RNPs by plotting them along a continuum of flexibility. As alluded to above, flexibility can refer to the functional organization of an RNP into modules that can function from different relative positions along an RNA scaffold, and it can also refer to the often-related characteristic of the RNA scaffold being physically dynamic and pliable (see [26]). These two forms of flexibility are likely to go hand-in-hand in most, if not all cases, but it is also important to be explicit when considering these physical organization principles, as we have tried to be in our studies.

#### 3.1.1. Defining the Limits: The Most Rigid RNPs

RNase P and the ribosome may be viewed as some of the most rigidly structured large RNP complexes characterized so far (Figure 2). It is these structured, famous RNPs that also dominate how most of us envision RNP physical organization. The ribosome’s highly structured nature is presumably correlated with the fact that it is amenable to X-ray crystallography [55,56], despite its massive size. The same can be said of RNase P, except that it is just not as large or ornately complex. However, the ribosome certainly does flex, and the large and small subunits disassociate, etc. Its physical dynamism is critical for translating mRNA into protein (nicely brought to life in Venki Ramakrishnan’s Nobel Prize lecture [57]). Thus, one can envision RNP complexes that could be even more structured than the ribosome if they did not also happen to be constrained to carry out enzymatic actions, which require dynamics to handle sequential enzymatic reaction steps of binding to substrates, coordinating transition states, releasing products, etc. Such an RNP could be the Vault complex, although its RNA composition appears to be far less than protein subunits. The Vault probably belongs in the class of “protein-determined RNP structures” (for classes based on relative RNA vs. protein composition in RNPs, see [54]).

#### 3.1.2. Defining the Limits: The Most Flexible RNPs

In terms of the most flexible large RNPs, it is interesting to consider what would be theoretically, as well as biologically, the most extreme case. In theory, single-stranded RNA would be most flexible (this is the schematic on the far right of Figure 2), yet in physiologically relevant aqueous environments, a large RNA cannot be expected to avoid base pairing, and even ssRNA will have base stacking and other levels of structure formation not caused by Watson–Crick, G•U, and noncanonical base pairings. So, in practical, biologically relevant terms, pre-mRNAs seem to fit this noncoding RNP category best (Figure 2), since they have not “coded” for a protein yet, are large, and are complexed with various proteins [58]. Although it may seem unconventional to consider a pre-mRNA bound to proteinaceous factors a noncoding RNP, it would be neglectful not to keep this vastly diverse population of RNA-protein complexes in mind. In the absence of knowing of great numbers of single-stranded RNAs that avoid intramolecular pairings, it may be, in practical terms, that highly helical RNAs, with little tertiary structure and littered with bulges and internal loops, which comprise hinges, are about the most physically flexible RNA one is apt to find in cells. Notably, TLC1 mostly fits this description (Figure 1).

However, like the ribosome, yeast telomerase is an enzyme, and I have mentioned above that it has structured elements within its conserved domains. These modules bind to specific proteins and coordinate what is presumably a complex and elegant “dance” with its core-enzyme partner TERT during rounds of template re-use in telomere synthesis. Thus, additional examples of such extremely flexible RNPs are apt to be discovered, and presumably the most flexible will be classes of lncRNAs that bind to proteins for the sole purpose of scaffolding them in a flexible fashion, presumably without the constraints of also coordinating enzymatic RNP dynamics.

Some of the most flexible RNAs in large RNPs are likely to be lncRNAs. Therefore, although lncRNAs are a broad classification, I have listed them to the right of yeast telomerase in Figure 2, despite their physical organization remaining largely uncharacterized. Some lncRNAs, such as HOTAIR, have, based on the example of TLC1, been proposed to provide flexible scaffolds for proteins [59,60]. Overall, it has been clear for some time that even human lncRNAs are rapidly evolving [61], which is one of the central hallmarks of the telomerase RNA paradigmatic flexible-scaffold RNA, as described in Section 2 above. Once more lncRNAs are evaluated in terms of biochemistry and mechanism, and these candidates and others will be vetted and, in turn, can be plotted onto the continuum of large RNP flexibility shown in Figure 2. Very recent results from using atomic force microscopy (AFM) and EM for human HOTAIR, for example, are beginning to show that this lncRNA is strikingly flexible, with many possible conformations observed in cells [39], consistent with it being a flexible scaffold.

Limited progress on lncRNA structure–function has been reported, although there have been a substantial number of perspectives published on the topic, given the level of interest to RNA biologists and relevance of lncRNAs to human health [54,62,63,64]. However, fundamentally, nearly all lncRNAs remain wholly undefined in terms of whether they have substantial overall tertiary structure, and even their secondary structures are largely only bioinformatically modeled, with a few notable exceptions [62,64]. The relative lack of tertiary structure determination may be largely because of lncRNAs being frequently devoid of overall reproducible 3D structure, which would inhibit them from yielding high-resolution results in gold-standard structural biology techniques such as crystallography and cryo-EM, but this can only be inferred from the paucity of publications reporting lncRNA 3D structure.

### 3.2. What TLC1 Flexible Scaffolding Implies for other Telomerase RNPs

In terms of what yeast telomerase can tell us about telomerase flexibility in other species, as pointed out in Section 2.5 above, there are some rapidly evolving parts of vertebrate telomerases, including humans. The “hypervariable” region in hTR secondary structure presumably helps to tether the essential CR4/5 domain TERT-binding interface to the catalytic core (where TERT also binds) in order to provide full function of the enzyme. This hypervariable stem is composed of two rapidly evolving regions that come together to pair in the secondary structure, and it seems to be one of the areas of the RNA that will be hardest to “pin down” in 3D models as structural biologists in the telomere field try to resolve a high-resolution vertebrate telomerase RNP structure [52,65,66]. It may well be that even if the hypervariable region can be captured in a structure, that at least for this region, its conformation would be one of many possible snapshots given its likely dynamic physical nature.

As for why telomerase in yeast is so flexible, and why it is rapidly evolving in sequence and size throughout the Tree of Life, this is a fascinating question, but the answer remains elusive. Presumably, there are facets of telomerase RNP biogenesis, assembly, catalytic mechanism, regulation, and more, which are beyond our understanding at this point, leaving such large questions too far “beyond the horizon” to be worthy of speculation just yet.

## 4. Conclusions

In summary, investigating yeast telomerase RNA has taught us about not just this particular RNP enzyme, but also about the organization of an emerging multitude of other RNA-based complexes that are also likely to be flexible. As the anatomy of other RNA-containing complexes is characterized, it will be useful to consider the impressive possible, and likely extant, physical and functional flexibility of RNA complex organization compared to TLC1. The proposed continuum of RNP complex flexibility in Figure 2 will hopefully help researchers to assess a given complex’s flexibility by plotting it relative to benchmark RNPs along the continuum. There remain far more examples of structured noncoding RNPs than ones demonstrated to be flexible, but this is apt to change substantially as the study of lncRNAs expands and as RNP research shifts increasingly towards understanding physical organization and functional mechanisms.

One of the best ways to identify a highly flexible RNA is based on a fast rate of sequence evolution and by noting resistance to structure being captured in 3D by X-ray crystallography or ensemble-averaging of cryo-EM data. However, even presumably flexible lncRNAs can be studied biophysically, such as by AFM, EM, and other single-molecule approaches. These approaches are likely to be more frequent as more researchers consider how the term “structure” tends to presuppose rigidity of macromolecular RNP complexes. This research will seek increasingly more than structural identification of three-dimensionally organized domains, but also biophysical characterization of dynamically flexible full-length RNAs. As we have shown in studying yeast telomerase RNA, when a large RNA is rapidly evolving overall, yet contains discrete islands of conserved sequence, this affords the ability to test for functions in vivo and in vitro upon altering the relative arrangement of the conserved modules along the transcript. Few studies of RNPs have been performed in this way, but this approach will become more widespread, as it should reveal far more RNAs that function as flexible scaffolds for proteins and other modular moieties.

## Figures and Tables

**Figure 1 molecules-25-02750-f001:**
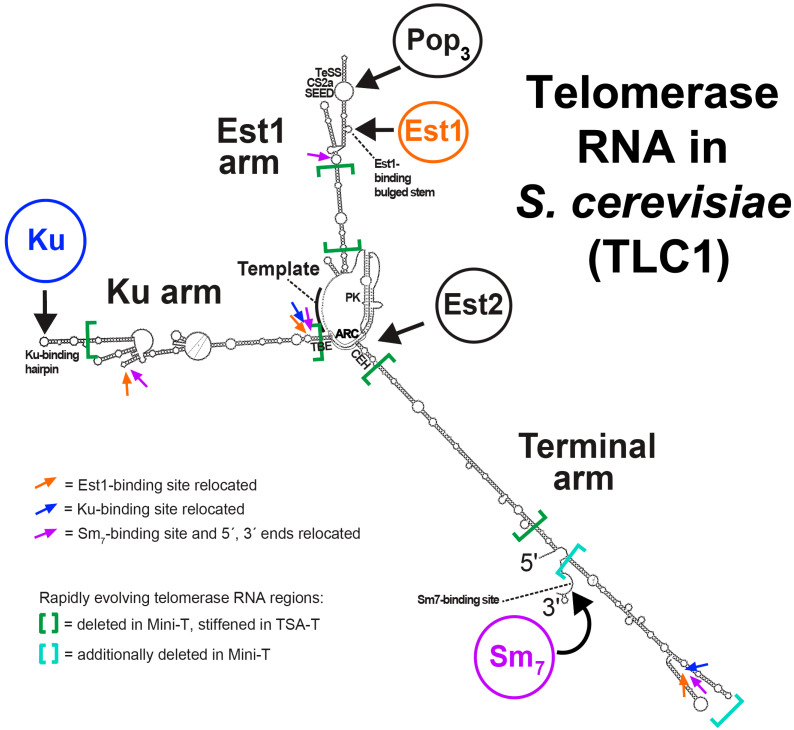
The *Saccharomyces cerevisiae* telomerase RNA: a paradigm for RNA as a flexible scaffold of a large RNP. Shown is the secondary structure model reported originally [13] along with the dynamic pseudoknot (PK) region in the central core, which was worked out in detail over the following decade [18,19,20,21,22]. Positions used for moving the binding sites for the Est1 [13], Ku [23], and Sm subunits (along with the RNA ends) [24] are indicated by colored arrows. Large green brackets indicate the rapidly evolving regions, which were deleted, in the case of making the miniaturized TLC1 allele Mini-T [25], or stiffened, in the case of triple-stiff-arm TLC1 (TSA-T) [26]. Thus, it is the regions outside of the brackets that are the more conserved modules that tend to bind to proteins at the tips of the arms, and coordinate catalysis with TERT (Est2) in the central core. Also indicated on the model are the conserved secondary structure elements, most of which have well-defined binding partners (e.g., the region above the Est1-binding bulged stem has been shown to bind to Pop1/6/7 (Pop_3_) complex [27,28]) or are part of the catalytic core. The function of the three-way junction at the tip of the terminal arm is an exception in that it is well-conserved (even with strong structural homology to the human telomerase RNA (hTR) CR4/5 domain, which binds to hTERT), yet it is dispensable in yeast [15,25] and has no known binding partner or function. ARC, area of required connectivity [29]. CEH, core-enclosing helix [30]. PK, pseudoknot [19]. TBE, template-boundary element [10,12].

**Figure 2 molecules-25-02750-f002:**
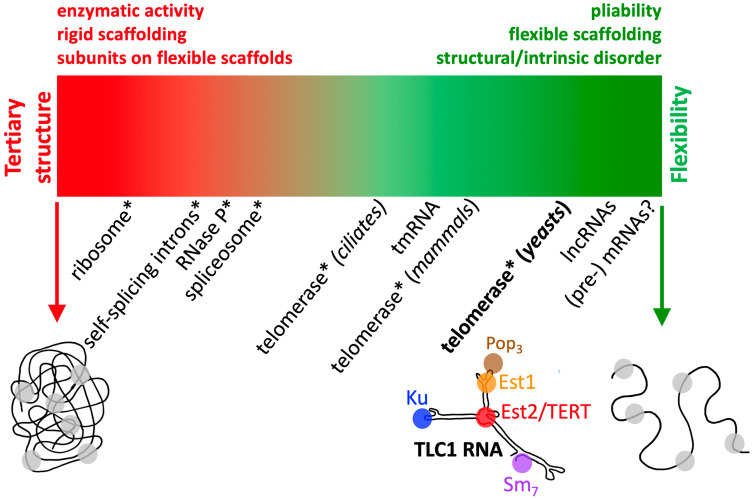
Physical organization of large RNA-protein complexes along a flexibility continuum. An extreme hypothetical structured and floppy RNP scaffolded by RNA is shown on each end of the continuum. Noncoding RNAs that bind to proteins in RNPs (in a broad sense of defining an RNP) are listed below the continuum based on where one might plot them on the flexibility spectrum. Telomerase RNA from yeast, TLC1, is shown as a schematic, since it is an archetype for flexible scaffolding RNAs in an RNP. In cases where an RNP complex is known to have enzymatic (typically at least partially also ribozyme) activity, the RNP is highlighted with an asterisk (*). Enzymatic activity tends to be a feature requiring structure, so most enzymatic RNPs are on the structured size of the spectrum; in this regard, yeast telomerase is exceptional, given it is a mixture of flexibility and modular structured domains (foremost among them its catalytic core, bound to TERT) along the presumably floppy RNA tethering scaffold that assembles and orchestrates the RNP’s functions. Furthermore, telomerase is shown several times along the continuum, since in different species its physical organization is strikingly different.

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
