# Peer review of "Yeast Telomerase RNA Flexibly Scaffolds Protein Subunits: Results and Repercussions"

_molecules, 2020, doi:10.3390/molecules25122750_

Round 1

Reviewer 1 Report

This is an interesting and up-to-date review of the prototypical flexible scaffold of yeast TLC1 RNA.  Comparisons are made to lnc RNAs but more discussion  of this would be helpful, perhaps a figure of Hotair or Xist or some lncRNAs of known structure.  The TLC1 RNA is thought to be physically flexible because of extensive mutational analysis.  Are there other methods to determine RNP flexibility?

1.  There is discussion of different yeast TLC1 structures.   A figure showing their different sizes and also showing common regions would be interesting. The truncated T's could be included.

2.  Sizes of mammalian TRs and yeast TRs shoud be mentioned early on (2.1) rather than later.  Maybe even included in new suggested figure above.  Is there any sequence conservation between yeast and mammals or other organisms?

3.  Section 2.3:  what are functions of Est1, Ku, Sm7?  Do they need to work together?

4.  Line 198:  mentioning difference in copy number here seems unnecessary since you later say that's not important.  Should be reworded, shortened.

5.  Fig. 1 key:  Est1 binding binding   (typo)

Minor points for consideration

6.  Line 27:  biochemists or biologists?

7.  Line 40:  RNA world:   Should be referenced

8.  Line 72:  RNAs' seems awkward.  How about RNA sequences?

9.  Line 81:  help solve

10.  Line 88:  a not the

11.  Line 121:  I prefer confirmation to validity

12.  Line 202:  avoiding

13.  Line 214:  Delete either

14.  Line 221:  Minority should be replaced by a fraction or something like that.

15.  Line 242:  Not clear if you're comparing to 450 or 150 nts.

16.  Line 328:  Delete a

Reviewer 2 Report

This is a well-written manuscript reviewing the research into yeast telomerase RNA and its impact on the larger field of RNAs that scaffold proteins. The review is written in a first-person, rather colloquial manner, which is refreshing and appropriate, considering the author’s major contributions to the field.  See a few small suggestions to further improve the manuscript below.

Comments:

  1. One element that could be expanded on is discussion of the known roles/activities of some of the essential protein components of yeast telomerase, such as Est1 and Ku, etc. While the review is focused on the RNA component, it helps to consider how the results of manipulating the RNA are viewed, in reference to the specific activities of the telomerase holoenzyme proteins.

  1. On occasion, past tense is used when present would be preferable, such as: Lines 71 and 73.

  1. On Line 35 “sometimes protein” seems a bit too RNA-centric. Also Line 40 may benefit saying “likely began with RNA”.

  1. Line 45, “execute ancient processes” is unclear when referring to a rather newly-evolved lncRNA.

  1. Figure 1 may benefit from colored highlights around the regions such as the CEH, TBE, etc., to clearly define them. Also, the nucleotide ranges in the figure legend would be helpful.

  1. Lines 111-113 are not a very clear statement. Is this trying to say that regions with predicted complex structure were those that had phenotypes when deleted?

  1. Section starting at Line 143 is title “Est1”, but most Est1 text appears above and the text in this section is more general.

  1. Section beginning with Line 150 continues a thought from a previous section, which is rather awkward.

  1. May be useful to define “circular permutation” (Line 157).

  1. Line 160: reference to co-transcriptional folding pathway is intriguing. Is it worth briefly elaborating?

  1. Line 210 (and above): TSA-T construct was not discussed as being less than fully fit.

  1. Line 329: mRNPs are frequently discussed in the literature, so this statement that it is “unconventional” to consider them may need re-phrasing.
  2. Line 380-381: Many RBPs specifically bind ssRNA only (most RRM-RNA co-crystal structures, for example). The presumption that all RNA-protein interactions require secondary structure or beyond should be re-considered.

  1. Locations of minor editing fixes needed:
  • Line 34
  • Line 296
  • Line 382
